# Clinically Significant *BRCA1* and *BRCA2* Germline Variants in Breast Cancer—A Single-Center Experience

**DOI:** 10.3390/cancers17010039

**Published:** 2024-12-26

**Authors:** Răzvan Mihail Pleșea, Anca-Lelia Riza, Ana Maria Ahmet, Ionuț Gavrilă, Andreea Mituț, Georgiana-Cristiana Camen, Cristian Virgil Lungulescu, Ștefania Dorobanțu, Adina Barbu, Andra Grigorescu, Cecil Sorin Mirea, Michael Schenker, Florin Burada, Ioana Streață

**Affiliations:** 1Regional Centre of Medical Genetics Dolj, Emergency County Hospital Craiova, 200642 Craiova, Romania; razvan.plesea@umfcv.ro (R.M.P.); anca.costache@umfcv.ro (A.-L.R.); andreea.crgm@gmail.com (A.M.); stefania.crgm@gmail.com (Ș.D.); adina.crgm@gmail.com (A.B.); florin.burada@umfcv.ro (F.B.); ioana.streata@umfcv.ro (I.S.); 2Laboratory of Human Genomics, University of Medicine and Pharmacy of Craiova, 200638 Craiova, Romania; andra.grigorescu97@gmail.com; 3Doctoral School, University of Medicine and Pharmacy of Craiova, 200349 Craiova, Romania; anaahmet@yahoo.com; 4Department of Radiology and Medical Imaging, Faculty of Medicine, University of Medicine and Pharmacy of Craiova, 200349 Craiova, Romania; georgiana.camen@umfcv.ro; 5Department of Medical Oncology, University of Medicine and Pharmacy of Craiova, 200349 Craiova, Romania; cristian.lungulescu@umfcv.ro; 6Department of Surgical Semiology, Faculty of Medicine, University of Medicine and Pharmacy of Craiova, 200349 Craiova, Romania; cecil.mirea@umfcv.ro; 7Department of Medical Oncology, Sfantul Nectarie Oncology Center, 200801 Dolj, Romania; michael.schenker@umfcv.ro

**Keywords:** *BRCA1*, *BRCA2*, breast cancer, genetic susceptibility

## Abstract

Pathogenic and likely pathogenic germline variants in the *BRCA1* and *BRCA2* genes play a pivotal role in breast cancer development and progression and can determine the optimal risk-reducing strategies and personalized case management for the carriers of such variants. Our study aimed to evaluate the carrier status in a group of 58 patients who were referred to our center for genetic testing of the two genes, as well as establish a set of correlations between their genotypes and their clinical–pathological features. The study revealed that 15.5% of the patients harbored pathogenic variants in either of the two genes and that carriers of the *BRCA1* pathogenic variants manifested a more aggressive tumor phenotype. These findings provide valuable insights that could be useful for the improvement in current national screening strategies and consolidate genetic testing as a valuable instrument in the personalized management of breast cancer.

## 1. Introduction

In 2020, approximately 2.3 million new BC cases were reported; this represents 11.7% of all cancer cases [1,2]. BC is the fifth leading cause of cancer-related deaths worldwide, with 685,000 deaths [1,2] despite the recent advances in personalized cancer therapy such as oncogene targeting, CAR-T, or gene therapy [3]. It is projected that by 2040, the incidence of newly diagnosed cases will rise to approximately 33.8% [1]. BC has surpassed lung cancer as the primary contributor to the rising global incidence of cancer in women, accounting for one in four cancer cases and one in six cancer-related deaths in women.

In Romania, BC is diagnosed annually in approximately 10,000 women and is responsible for approximately 3300 deaths among females [4]. Additionally, there has been a consistent upward trend recorded of malignant breast tumors in women, rising from 56,251 in 2013 to 73,021 in 2020 [5,6,7]. While the reported incidence rate of BC in Romania is approximately two times lower than that of the European Union, the mortality rate is closer to the European average (30.7%_000_). In 2020, in the southwestern region of the country, the mortality rate due to malignant breast tumors was around 23%_000_ in Dolj county, reaching approximately 27%_000_. Despite regional differences, the lower survival rates may be attributed to late-stage diagnosis as a common occurrence in Romania [8].

Complex BC etiology and pathogeny contribute to these concerning statistics. Susceptibility to this condition can be influenced by modifiable and non-modifiable risk factors, including genetic factors. Gene expression analysis using microarrays allowed for the classification of breast tumors into molecular subtypes which first displayed an obvious specificity in their gene expression markers and were then identified using immunohistochemistry; they include the luminal A, luminal B, Her2-positive, and triple-negative subtypes [9,10,11,12,13,14,15,16,17,18,19,20,21,22,23]. They proved to reveal significant differences in their grade and prognosis (prediction of disease-free survival and overall survival), which resulted in the designing of clearly different therapeutic strategies for each subtype. Thus, the luminal A subtype is usually a low-grade proliferation that benefits from endocrine therapy and has a highly favorable prognosis. The luminal B subtype is moderately differentiated, with differentiated therapy algorithms depending on the Her2 positivity and with an intermediate prognosis. The Her2-positive subtype encompasses high-grade tumors with poor prognosis which still benefit from targeted therapy. Finally, the triple-negative subtype includes high-grade tumors with morphological, molecular, and clinical heterogeneity, as well as with the worst prognosis, requiring complex, combined therapeutic algorithms [24,25,26,27,28,29,30,31,32].

In addition, gene expression analysis identified variations in genes such as *BRCA1* and *BRCA2*, known tumor suppressor genes with a role in DNA repair, or in the genes that interact with *BRCA1* and *BRCA2* [33]. A variation which can be inherited is called a germline variation; on the other hand, somatic variations can be acquired in isolated tissues due to a combination of genetic, environmental, and lifestyle factors. One of the processes through which epithelial malignancies progress to a higher phenotype is type 3 epithelial–mesenchymal transition, a process highlighted in many tumors, including kidney, bladder, and even breast proliferations [34,35,36]. In breast tumors, this higher phenotype defines the basal-like cells of triple-negative tumors [34].

Germline variants in the *BRCA1* and *BRCA2* genes have significant implications, as they are associated with a higher likelihood of developing certain types of BC. For *BRCA1* carriers, the estimated risk of developing BC is 60% for one breast and 83% for both breasts. Similarly, for *BRCA2* carriers, the cumulative risk is estimated at 55% for one breast and 62% for both breasts [4,37,38,39]. They are also associated with a higher likelihood of developing particular types of BC. For instance, *BRCA1* is tightly linked with many of the molecules involved in the epithelial–mesenchymal transition process, with this relationship being able to determine the appearance of aggressive tumor phenotypes like triple-negative variants [40,41,42]. Moreover, the presence of pathogenic or likely pathogenic variants in the *BRCA1* and *BRCA2* genes is strongly associated not only with breast tumors but also with ovarian (OC), prostatic, and pancreatic cancers. The mutational patterns observed in these genes include frameshift variants, nonsense and missense mutations that disrupt protein function, splice site mutations leading to protein truncation, and large rearrangements [43,44,45].

Lastly, both germline and somatic gene mutations are also related to the therapeutical step in the management of BC. The studies carried out over time have shown, on the one hand, that particular germline gene mutations could benefit from particular therapeutic schemes. For instance, in tumors with pathogenic *BRCA1* mutations associated with hormone receptor positivity, hormone therapy (tamoxifen and aromatase inhibitors) delays tumor progression, reduces risk significantly, and prevents the onset of contralateral tumors [46,47]. In turn, in hormone receptor-deficient tumors, the combination PARP inhibitors and chemotherapy/immunotherapy can increase the killing effect of *BRCA1* germline mutation [48,49]. Finally, another interesting observation of the researchers was that, in mutated *BRCA1/2* tumors, secondary mutations of *BRCA1/2* were identified that were associated with an acquired resistance to previously efficient drugs [50,51,52,53].

Taking the above issues into consideration, through the present study, we aimed to assess the mutational status of the *BRCA1* and *BRCA2* genes in a cohort of individuals with malignant breast tumors in the southwest region of Romania.

## 2. Materials and Methods

The study group consisted of 58 patients, with a personal history of BC, who were referred to the Regional Centre for Medical Genetics (RCMG) Dolj, Craiova, Romania, for genetic investigation and counseling during a three-year period. The inclusion criteria for the study group were female patients with genetic testing for germline PVs or LPVs in the two *BRCA* genes.

The initial group of cases was divided into two main groups according to the results of genetic investigation, namely *BRCA* non-carriers and *BRCA* carriers. Secondly, the *BRCA* carrier group was divided, according to the mutation type, into *BRCA1* carriers and *BRCA2* carriers.

The selection criteria for genetic testing followed the guidelines and standards established by the National Comprehensive Cancer Network (NCCN) and the European Society for Medical Oncology (ESMO) [45,54]. These criteria encompassed the specific indications and risk parameters related to oncological conditions.

Collected data included age, personal and family history, clinical phenotypes, diagnostic procedure, histogenetic type of the lesions, histopathological (HP) diagnosis (if carried out), and molecular classification of lesions (if carried out).

The appropriate pre- and post-test genetic counseling was offered to all subjects, adhering to best practice protocols.

Data concerning breast lesions were obtained by clinical examination, biopsies, and surgical procedures with histopathological examination.

For the histopathological assessment, tissue samples were processed using the classical HP technique (formalin fixation and paraffin embedding). The hematoxylin–eosin stain was used for the histopathological assessment, which was performed in accordance with the last WHO classification of breast tumors [55]. The immunohistochemical three-stage indirect Avidin–Biotin Peroxidase complex method was used for the molecular classification of lesions which was carried out in accordance with the works of Perou et al. [9] and Sørlie et al. [10,11,12], as updated by Tsang et al. [24]. The antibodies used and their significance are listed in Table 1.

Index case germline genetic testing was performed on EDTA venous blood using next-generation sequencing (NGS). The Ampliseq for Illumina (Illumina, Inc., San Diego, CA, USA) *BRCA* panel with Illumina Rapid Capture library preparation kit was used. Paired end 2 × 150 bp reads on the Illumina NetSeq550 IVD sequencing platform with at least median 100× coverage were mapped to GRCh37 using the iGenomes resource bundle, and pushed through the nf-core/sarek 2.7.1 pipeline. Variants with a depth of over 20× were considered for diagnosis. Additionally, coverage of the *BRCA1* and *BRCA2* genes was manually investigated. Situationally, capillary sequencing was used to obtain the full coverage of exons in the genes of interest. Targeted testing for the identified variants among family members of the index cases was performed using the ABI3730 capillary sequencing platform from Applied Biosystems™ (Thermo Fisher Scientific Inc., Waltham, MA, USA). The MutationSurveyor^®^ DNA Variant Analysis Software v.5 (Softgenetics, State College, PA, USA) was used for data analysis.

We identified the pathogenic/likely pathogenic variants and classified them according to the ACMG guidelines [56,57]. This classification is applicable to variants in all Mendelian genes and comprises a five-tier system of classification for variants relevant to Mendelian disease. The germline variants identified were annotated using ENSEMBL variant effect predictor (VEP) [58], with several plugins for predictive scores; online aggregate databases such as OMIM [59], ClinVar [60], Varsome [61] were also consulted. Segregation data, where available, were used for ACMG-compliant variant classification [56,57].

## 3. Results

The patients with a modified status of the *BRCA* genes represented less than 20% (15.5% precisely) of the entire group of patients tested.

The clinical pathological findings of the patients are summarized in Table 2.

### 3.1. Age

A comparative analysis of the patients’ ages in the two subgroups revealed some small differences between the non-carriers and carriers; the mean age of the former was higher, with one year more than that of the latter, and, consequently, more patients belonging to the carrier group were younger than 40 years of age (Table 2). In other words, patients carrying tumors with *BRCA* mutations were discovered a bit earlier than those without mutations.

### 3.2. Family History

The presence of BC in the family histories of the investigated patients was different in the two subgroups. On the one hand, almost two-thirds of the patients with no *BRCA* PVs had no relatives (grandmother, mother, aunt, sister, cousin, daughter) with BC, whereas more than half of the carrier patients had relatives with breast malignancies. On the other hand, *BRCA* PVs were present in almost 21% of patients with a family history of BC but in only 11.7% of patients without a family history of BC. However, the statistical tests did not validate these differences as significant (Table 2).

### 3.3. Diagnostic Procedure

Considering the type of diagnostic procedure used to determine the patients that were suitable for further molecular assessment, three distinct categories were observed. The first category consisted of 21 patients that carried out the molecular investigation following a clinical examination which revealed a mammary nodule. Most of these patients were *BRCA* non-carriers, except for one 28-year-old patient who expressed a *BRCA1* mutation. The second category included only 10 patients who were referred to the Regional Centre for Medical Genetics after a clinical examination which revealed a mammary nodule followed by a guided biopsy of the nodular lesion. Almost two-thirds of these cases were *BRCA* non-carriers, with the rest of the cases proving to be *BRCA* carriers and representing almost half of the patients with a *BRCA* mutation. It should be noted, however, that the histopathological result of the biopsies revealed a malignant proliferation within the clinically detected nodule/nodules in all these cases. The third category included patients with an established diagnosis of breast malignancy; most of them proved to be non-carriers but the rest of them represented the other almost half of the cases harboring *BRCA* mutations.

The group of non-carriers mainly included either patients that underwent a surgical procedure and were diagnosed with a type of BC or patients with a clinical diagnosis of a mammary nodule. In turn, the group of carriers included mostly patients with an established histopathological diagnosis either by biopsy or after a surgical procedure. These differences were, however, not validated as significant from the statistical point of view (Table 2).

### 3.4. Histogenetic Types

There were differences between the two groups concerning the determination of the histogenetic type of mammary lesions. Thus, almost one-third of the non-carrier cases had no specification of the histogenic type of nodular lesions and that was because all these cases belonged to the group of patients with a clinical examination only. In 10% of the non-carrier cases, the supposed diagnosis was of fibro-cystic changes. However, from a histopathological point of view, more than half of the cases were diagnosed as malignancies of the mammary gland.

In turn, with one exception, all carrier cases were diagnosed as breast carcinomas. However, these differences were not validated from the statistical point of view as significant (Table 2).

### 3.5. Histopathological Diagnosis

Moving beyond the assessment of histogenetic type, HP profile determination showed different situations in the two groups of patients. In more than one-third of the non-carrier cases, HP evaluation did not follow the genetic testing, as patients were only concerned with assessing their risk of developing BC. Further, in less than 10% of the cases, the histological examination revealed only fibro-cystic changes in the mammary parenchyma. All the other cases were malignant epithelial proliferations. Except for one case of mucinous carcinoma, as well as one case of “in situ” proliferation, all the other tumors proved to be invasive carcinomas that were mainly of the ductal type.

Almost the same situation was observed in the group with *BRCA* PVs, where, with the exception of one case without HP evaluation, all the cases turned out to be invasive carcinomas and almost all of the ductal type.

### 3.6. Molecular Classification

Immunohistochemical assays for the molecular classification of breast lesions, the only parameter crucial for the design of a therapeutical strategy, were not carried out in all the patients. In the non-carrier group, eighteen patients only had a clinical examination, following which they referred to the RCMG for genetic testing; the other five patients, after a clinical examination, had a sample of breast tissue taken which was not subjected to immunohistochemical assays after the standard histological examination. In more than half of the rest of the cases in this group, immunohistochemical assessments revealed that many of the tumors (12 cases—46%) were luminal A-type carcinomas, followed by luminal B-type carcinomas (14.3%) and Her2-positive tumors (10.2%). In the BRCA carrier group, more than half of the tumors (five cases—55%) were triple-negative tumors but only two of these had a high Ki67 index. Two of the cases were luminal-type malignancies and, in two cases, immunohistochemical assays were not carried out (Table 2 and Figure 1).

This striking difference between the two groups was validated as highly significant by the statistical tests (*p* value of χ^2^ test was 0.007).

### 3.7. Assessment of BRCA Variants

Deleterious germline *BRCA1* or *BRCA2* variants (pathogenic (P)/likely pathogenic (LP) variants) were identified in 9 out of 58 (15.5%) patients (Table 2). Variants of uncertain significance (VUS) were not found in our cohort. *BRCA1/2*-related BC had similar pathological characteristics with regards to the sporadic tumors in terms of histological grade and lymph node involvement. Table 3 presents the pathogenic or likely pathogenic *BRCA1/BRCA2* variants that we identified in our study group.

The *BRCA1* variant *NM_007300.4:c.5329dup* was found in three females, aged 58, 45, and 28. The 58-year-old female was diagnosed at age 49 and underwent a radical mastectomy for left BC at the same age. She has a daughter with BC. This frameshift null variant introduces an early termination signal, potentially producing a truncated protein or leading to the absence of a protein through a mechanism known as nonsense-mediated decay (NMD). Both scenarios are recognized pathways to disease [78]. The impacted exon affects a single functional domain, specifically the ‘BRCT 2’ domain as denoted by UniProt’s *BRCA1*_HUMAN protein annotation. Within this exon, 98 pathogenic variants have been reported, while the area that would be truncated by the mutation contains 317 pathogenic variants.

Another *BRCA1* variant, *NM_007300:c.843_846delCTCA*, introduces an early termination signal, also producing NMD. *c843_846del* was found in our study in two female patients, with onset at the ages of 35 and 47, respectively. In both cases, the invasive ductal carcinoma was PR, ER, and HER2 negative. Only one of the cases had a positive family history of tumors; her sister had ovarian cancer, her mother gastric cancer, and her brother bladder cancer.

We observed the *BRCA1* variant *NM_007300.4:c.5093_5096del* in a 63-year-old patient with BC and a negative family history. This frameshift null variant is anticipated to lead to nonsense-mediated mRNA decay (NMD). The loss-of-function effect is a well-established cause of disease, supported by 3443 previously reported pathogenic loss-of-function variants for this gene. The affected exon interacts with a specific functional domain, namely the ‘BRCT 1’ domain, as identified in the UniProt entry for the human BRCA1 protein. Within this particular exon, there are 72 known pathogenic variants, and the region that would be truncated due to this frameshift contains 624 pathogenic variants.

The *BRCA2* variant *NM_000059.4:c.2471T>G* was identified in a 61 year-old patient that was diagnosed with bilateral breast ductal carcinoma at the age of 44. Her family has a rich clinical history, with her mother and daughter having BC and her sister being diagnosed with ovarian cancer.

*BRCA2 NM_000059.4:c.5576_5579delTTAA* has been cited in breast and ovarian cancer cases [74,75,77]. The variant has not been reported in general population-based databases [82]. The deletion of four nucleotides leads to a frameshift that likely causes a truncated protein or a complete absence of the transcript. We identified the PV in a 42-year-old woman without a history of breast or ovarian neoplasia.

The *BRCA2* variant *NM_000059.3:c.8331+1G>A* was identified in a 52-year-old patient that was diagnosed with bilateral BC at the ages of 39, with lobular carcinoma in the right breast, and 47, with ductal carcinoma in the left breast; both were PR and ER positive and HER2 negative.

### 3.8. Clinical and Pathological Differences Between the Two Carrier Subgroups

We further analyzed both groups of *BRCA* PV carriers to see if they showed different clinical–morphological profiles (Table 4). The patients carrying *BRCA1* PVs represented two-thirds of the entire group of carriers.

*Age*. The comparative analysis of the patients’ ages in the two subgroups revealed that one-third of the *BRCA1* carriers were younger than 40 years whereas all the *BRCA2* carriers were older than 40 years (Table 4).

*Family history of breast cancer*. We observed that two-thirds of the *BRCA1* carriers had breast malignancies in their family history, whereas two-thirds of *BRCA2* carriers had no breast malignancies in their family history.

*Diagnostic procedure*. The tumors of *BRCA1* carriers were diagnosed more frequently by clinical examination and biopsy while the tumors of *BRCA2* carriers were diagnosed more often by a surgical procedure followed by histopathological examination.

*Histogenetic type*. Tumors examined were practically malignancies in both subgroups of carriers.

*Histopathological diagnosis*. All the tumors investigated were invasive, but the tumors with *BRCA1* PVs were all ductal-type proliferations, whereas one of the three tumors with *BRCA2* PVs was of lobular type.

*Molecular classification*. Except for one case, which lacked immunohistochemical evaluation, nearly all the other tumors (two-thirds) in the subgroup of *BRCA1* carriers were of the triple-negative subtype. In turn, two-thirds of the subgroup of *BRCA2* carriers had luminal A-type tumors (estrogen receptor positive and low Ki67 index).

However, all these differences were not pronounced enough for the statistical tests to validate them.

### 3.9. Corelations Between Molecular Phenotypes and Genetic Variants

Another step in our analysis was to check if there was any correlation between the molecular profile and the presence or absence of different variants of BRCA mutations. In this respect, we took into consideration only those cases with two investigations performed. Thus, almost half of the non-carrier patients had luminal A-type tumors and, to a lesser extent, luminal B-type tumors (26.9%). Poorly differentiated tumors (Her2-positive and triple-negative subtypes) were present in the same percentage, with more than twice the prevalence of Her2-positive malignancies.

A total of 80% of the *BRCA1* carriers had triple-negative tumors, whereas two-thirds of the *BRCA2* carriers had luminal A tumors.

We can summarize that, in our study, non-carrier patients had the lowest aggressive tumors, while *BRCA1* carriers had the most aggressive tumors. *BRCA2* carriers were placed in an intermediate position (Table 5).

## 4. Discussions

Human *BRCA1* and *BRCA2* genes encode proteins that are crucial for the repair of double-stranded DNA breaks through homologous recombination [69]. Germline mutations in these genes lead to impaired DNA repair, resulting in genomic instability and susceptibility to cancer, notably breast and ovarian cancers.

Mutations in these genes can be inherited, leading to a significantly higher risk of developing cancer in carriers. Routine molecular profiling for deleterious germline variants in *BRCA1* and *BRCA2* has become an integral component of the diagnostic and therapeutic approach to hereditary breast and ovarian neoplasms. Individuals with a family history of breast or ovarian cancer are often encouraged to consider testing.

The penetrance of *BRCA* mutations is high, meaning that individuals carrying a deleterious mutation have a significantly increased lifetime risk of developing cancer compared to the general population. For instance, women with a *BRCA1* mutation have a lifetime risk of 55–72% of developing BC and 39–44% of developing ovarian cancer [83].

### 4.1. Carrier Profiles of the Two BRCA Genes in Our Study

Each of the *BRCA* PV carrier groups revealed a distinct profile. Thus, in our study, the *BRCA1* carriers were mostly over 40 years old, usually with a family history of BC. They were often diagnosed with a breast lesion by a clinical examination coupled with a biopsy. The breast lesion was almost always malignant, namely an invasive carcinoma that was either ductal or lobular and was usually a triple-negative tumor.

The *BRCA2* carriers were all older than 40 years, with no family history of BC. They were often diagnosed with a breast lesion by a clinical examination coupled with a surgical procedure followed by a histopathological examination. The breast lesion was always malignant, namely an invasive carcinoma that was either ductal or lobular and was usually a luminal A-type tumor. These differences between the two subgroups of *BRCA* carriers, together with the differences described previously between the *BRCA* carriers and non-carriers, are in concordance with the literature data, even though many of the important tested correlations (e.g., age, family history, histopathological diagnosis) were only trend-like in our study.

This means that, in general, *BRCA* carriers are more prone to developing BC than *BRCA*-negative patients; when they do develop BC, the tumors are of a higher grade and more aggressive, with a higher recurrence risk score and a worse survival rate. Further, *BRCA1* carriers are more likely to develop more aggressive BCs, with a worse prognosis and at an earlier age than *BRCA2* carriers [39,40,42,66,67,84,85,86,87,88,89,90,91,92]. The fact that our observations only have trend-like values can be explained by the small size of the study group, this limit being determined by the fact that our center is a very new one, and the study is one of the first attempts to present our activities.

### 4.2. BRCA1 and BRCA2 Mutational Status in Romania

There have been several studies carried out in Romania to evaluate the mutational status of *BRCA1/2* in the context of both BC [66,93,94,95,96] and combined breast and ovarian cancer [4,6,7]. The study groups consisted mostly of patients from the northwestern [4,66,93,94,95] and northeastern [6,7] regions of the country, and only one from the southern/southeastern region [96]. Throughout these studies, *BRCA1* variants were more prevalent than *BRCA2*, as was the case in our group. Some of the studies, although performed on larger groups and on patients with BC or OC, only identified four variants of uncertain significance (VUS), two of which were in *BRCA1* and the other two in *BRCA2* [4,66,94]. We did not report any VUS in our study mainly due to the smaller sample size given the more restrictive inclusion criteria.

The most frequently found variants were the *c.3607C>T* (p.Arg1203Ter) in the *BRCA1* gene, and *c.9371A>T (p.Asn3124Ile)* in the *BRCA2* gene, respectively. Interestingly, the *c.3607C>T (p.Arg1203Ter)* variant was not found within our study, although it has a high frequency in the northwestern and northeastern regions of Romania [4,66,94,95] and has also been frequently reported in the southern and southeastern regions of the European continent [97].

Also, *BRCA2 c.9371A>T* was not reported in our group despite it being previously reported in the Romanian population. This variant was reported in twelve cases in one study, indicating that it is a prevalent mutation among Romanian breast and ovarian cancer patients [4]. Additionally, another report highlighted it as the most common pathogenic variant described in the Romanian population, with seven cases of BC and six cases of ovarian cancer [94]. Moreover, a study in 2022 that included 250 women with BC and 240 with ovarian cancer undergoing germline molecular testing showed *c.9371A>T* to be one of the most common variants identified for *BRCA2* [95].

The *BRCA1 c.843_846delCTCA* has been reported in the Romanian population, with the following two cases identified: one in a patient with BC and one in a patient with ovarian cancer [66]. These occurrences suggest the presence of this mutation among the Romanian population affected by these cancers; however, the data do not specify an incidence rate in the general population.

*BRCA1 c.5329dup* is one of the more common *BRCA1* PVs observed in Romanian patients with breast and ovarian cancer. In a study that evaluated women of Romanian ethnicity, this mutation was identified 17 times among patients with these cancers. It was the second most prevalent variant after the *c.3607C>T* mutation in *BRCA1*. This suggests that *NM_007300.4:c.5329dup* is a significant PV within the Romanian population with breast and ovarian cancer. Although it is described as a founder mutation in some populations [94], an earlier study in the northeastern region of Romania suggested that it would not have a recurrent or founder effect in our country [7], with further studies being required on larger patient groups to sustain or infirm this hypothesis.

*BRCA1 c.5093_5096del* is a rare occurrence in the Romanian population, being reported in two patients within two different studies [94,95]. However, it has been reported in the literature in three patients within a Tunisian population with early-onset hereditary breast and ovarian cancer syndrome (HBOC) [69] and other patients in the Middle Eastern, North African, and South European countries [97].

We are reporting the *BRCA2 c.2471T>G* variant for the first time in Romania as it has not been reported in any of the existing studies within our country, probably due to the low addressability of genetic screening tests in the population. There are also only a few publications in the specialized literature where it has been mentioned, the most notable of which being a 2018 worldwide study on 29,700 families that harbor *BRCA* mutations [67].

The *BRCA2 c.8331+1G>A* intronic splice-site variant is also novel within the Romanian population and is one of the few that has frequently been reported in male BC patients [79,80,81].

The *c.5576_5579del (p.Ile1859fs)* variant in the *BRCA2* gene is only reported once in the Romanian specialized literature in one patient with BC in the northwestern region of the country [95].

These findings suggest that the variants reported in the current study are significant variants in the context of Romanian BC cases.

### 4.3. BRCA1 and BRCA2 Testing and Strategies

There are several benefits of the genetic screening of *BRCA* variants, including the following: (1) early detection through enhanced monitoring and surveillance follow-up plans in carriers; (2) preventive surgical intervention in carriers at high risk; and (3) establishment of therapeutic strategy. Recent randomized phase III trials have demonstrated the efficacy of therapeutic regimens involving platinum salts and PARP inhibitors specifically targeting certain germline mutations in patients with advanced BC [93,98].

Determining *BRCA* mutational status in patients with breast or ovarian malignancies can profoundly influence clinical decision-making, impacting both disease-free survival and overall prognosis. In recognition of its pivotal role, multiple international and national scientific consortia have promulgated various clinical management guidelines. These delineate both the surgical interventions and chemotherapeutic regimens optimized for prophylactic measures and therapeutic modalities, contingent upon the specific BRCA mutational profile.

Case management protocols for *BRCA*-associated BC syndrome are formulated based on a comprehensive understanding of the precocious manifestation of the disease, the augmented susceptibility to ovarian malignancies, and the propensity for male mammary carcinogenesis in those with a *BRCA1/2* PV. Emphasizing the imperative nature of early detection in individuals harboring the *BRCA* PV, it is quintessential for the timely identification of neoplastic transformations.

Individuals with a strong family history suggestive of HBOC are the primary candidates for genetic testing. This includes families with multiple cases of early-onset breast or ovarian cancer, bilateral BC, male BC, or combinations of other *BRCA*-associated cancers. Once a PV is identified in an index case, the cascade testing of at-risk relatives is essential for identifying other carriers who may benefit from enhanced surveillance or risk-reducing strategies.

In individuals with a hereditary predisposition to breast and ovarian cancer, specifically carriers of the *BRCA1/2* PVs/LPVs, early and intensive screening is of paramount importance due to the early age of disease onset. Protocols recommend breast awareness training from 18 years old, clinical breast examinations bi-annually from 25 years old, and tailored imaging schedules often starting in the mid-twenties, particularly with an MRI which has shown higher sensitivity compared to mammography. Data suggest that mammography might not be as effective in younger women due to factors like breast tissue density and rapidly growing tumors. MRI not only detects early-stage tumors with higher sensitivity but also reduces the radiation exposure risks associated with mammography. For optimal cancer risk management, both mammography and MRI are crucial, especially given the increasing evidence of MRI’s sensitivity in detecting tumors in *BRCA* carriers [66,67,68].

However, the specific intervals and imaging modalities remain a subject of ongoing research. Post-test counseling should extensively discuss risk-reducing surgical options and their implications. Ovarian cancer screening in high-risk women suggests potential earlier detection but definitive survival impacts are still under investigation. Men with the *BRCA* variants are also advised to start undergoing breast and prostate cancer screenings from specific ages. Ultimately, comprehensive *BRCA* screening and surveillance, combined with ongoing research, are critical to effectively manage and detect cancers early in individuals with a higher genetic risk.

The accessibility and coverage of *BRCA* testing are subject to healthcare policies and can vary depending on geographical location and healthcare systems, necessitating the consideration of healthcare equity in implementing testing strategies. While there has been a consistent rise in BC incidence in Romania over recent years, *BRCA* mutation testing remains largely inaccessible to medical professionals. As stated above, the RCMG Dolj is part of a recently formed national public network of regional medical genetics centers. Even though these centers have been established in the main administrative regions of the country, the addressability and accessibility of genetic screening tests are poor mainly due to increased costs in spite of the low income rates, especially in the rural areas, as well as the lack of national programs and government financing that could partially or totally subsidize these costs. Also, patients in the rural areas are more likely to be referred to municipal hospitals rather than regional ones, where the RCMGs are set in, for logistical and financial reasons. As a result, the range of *BRCA1* and *BRCA2* mutations in both sporadic and familial BC cases within our population is not fully characterized. This study contributes significant findings regarding a cohort of Romanian patients from the southwestern part of the country who underwent NGS *BRCA1/2* panel testing, among the few reported to date [66,93,94,95].

The scientific rationale for *BRCA1* and *BRCA2* testing is multifaceted, incorporating genetic epidemiology, molecular oncology, molecular pathology, guideline-driven clinical practice, therapeutic advancements, and ethical considerations. The decision for testing should be individualized, considering the person’s risk, family history, and preference, and should be conducted within a framework of comprehensive genetic counseling and informed consent.

Well-designed longitudinal outcome studies are also needed to clarify the prognostic outlook for patients with BC harboring germline *BRCA* PVs/LPVs at all disease stages [99] and establish how these would affect tumor microenvironment and potential novel immune therapies and treatment protocols using the latest tools in sequencing technologies such as targeted single-cell sequencing or long-read sequencing expanded by broader functional studies [100,101].

## 5. Conclusions

Our study, although conducted on a reduced number of cases coming from the southwestern region of the country, revealed that a significant percentage of the tested tumors carried pathogenic germline variants. The spectrum and frequencies of the germline variants in the *BRCA1/2* genes mirrored those described in the literature, and the *BRCA1* pathogenic variants were associated with the aggressive phenotypes of malignant proliferations. Therefore, *BRCA1/2* testing or broader genetic panels could be more efficiently popularized and implemented in cost-effective screening and risk-reducing strategies, contributing to the genetic epidemiology of breast cancer, enforcing its management both at a regional and national level, and providing optimal therapeutic options for patients harboring germline PVs, given the current availability of personalized therapy for these variants such as PARP inhibitors.

## Figures and Tables

**Figure 1 cancers-17-00039-f001:**
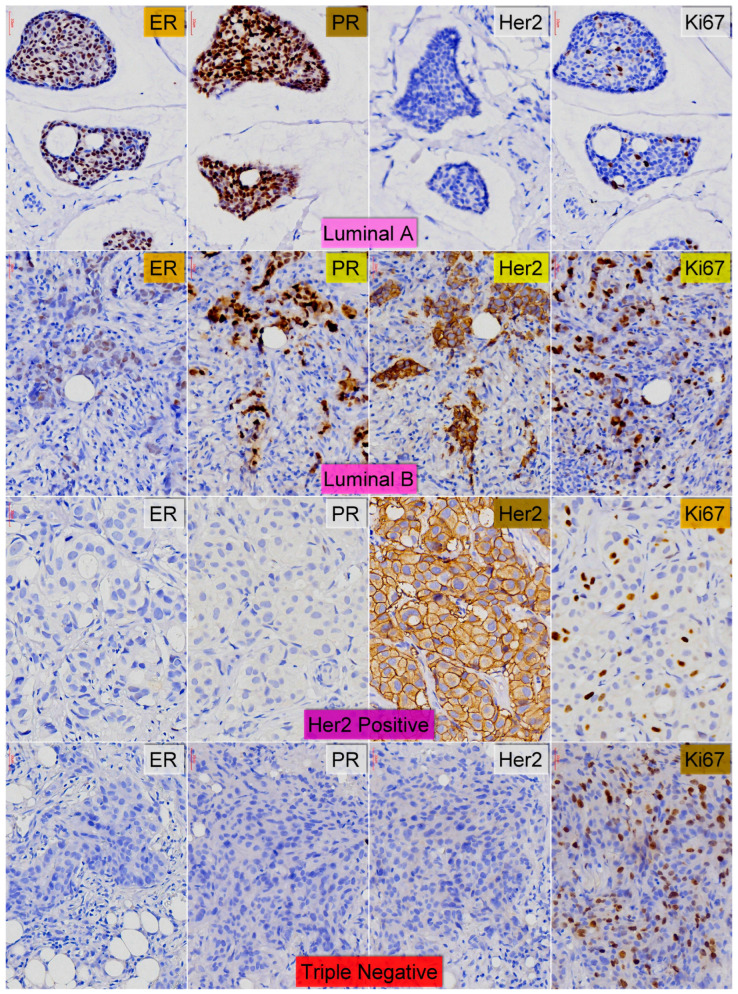
Molecular types of assessed breast carcinomas—Line I—Luminal A type (ER ≥ 1%; PR ≥ 20%; HER2 ≤ 10%; Ki-67 < 14%); Line II—Luminal B type+HER2+ (ER ≥ 1%; PR variable; HER2 > 10%; Ki-67 > 20%); Line III—HER2 enriched (ER < 1%; PR < 20%; HER2 > 10%; Ki-67 > 20%); Line IV—Triple-Negative (ER < 1%; PR < 20%; HER2 ≤ 10%; Ki-67 > 30%); Ob-X40 (all pictures).

**Table 1 cancers-17-00039-t001:** Antibodies used in the study.

Antibody	M/P	Clone	Source	Specificity	Significance	Dilution
ER	M	1D5	DAKO	Semiquantitative ER nuclear detection in BC	+—≥1%	1:50
PR	M	PgR636	DAKO	Semiquantitative PR nuclear detection in BC.	+—≥20%	1:50
HER2	M	4B5	DAKO	Membrane expression HER2 in BC	+—>10%	1:500
Ki-67	M	MIB-1	DAKO	Nuclear protein that is associated with and may be necessary for cellular proliferation Cellular marker for proliferation (Ki-67 Index)	Low—<14%	1:10
					Moderate—<20%	
					High—>20%	

Legend: BC = Breast cancer; ER = Estrogen Receptor; HER-2 = Receptor tyrosine kinase erbB-2; Ki-67 = Antigen Kiel 67; PR = Progesterone Receptor; M = Monoclonal; P = Polyclonal.

**Table 2 cancers-17-00039-t002:** Clinical pathological profiles of studied cases, segregated by mutational status in the two *BRCA* genes.

Groups			Non-Carriers		*BRCA* Carriers		Total
**Variables**	**Cases**		**49**		**9**		**58** **100%**
			85.5%		**15.5%**		
**Age**	** *≤40 years* **		7		2		**χ^2^ Test “*p*” value** 0.6495
	** *>40 years* **		**39**		**7**		
	** *NOS* **		3		0		
	** *Mean* **		51.6		50.3		
	**Two-sample *t*-test ”*p*” value** = 0.788						
**Family history for** **breast cancer**	** *Yes* **		19		**5**		0.3475
	** *No* **		**30**		4		
**Diagnostic procedure**	** *CL* **		18		1		0.1086
	** *CL + BIO* **		8		**4**		
	** *OP + HP* **		23		**4**		
**Histogenetic type**	** *NOS* **		16		1		0.187
	** *F-C Ch* **		5		0		
	** *M* **		**28**		**8**		
**Histopathological** **diagnosis**	** *NCO* **		18		1		0.494
	** *NN* **		3		0		
	** *Malignancies* **	** *DCIS* **	**28**	1	8	0	
		** *IDC* **		**19**		**7**	
		** *ILC* **		**5**		**1**	
		** *Mixed* **		**2**		0	
		** *MUC* **		1		0	
**Molecular** **classification**	** *NCO* **		23		2		**0.0034**
	** *L-A* **		**12**		1		
	** *L-B* **		7		1		
	** *HER2+* **		5		0		
	** *T-N* **		2		**5**		

Legend: BIO = Biopsy; CL = Clinical diagnosis; DCIS = Ductal Carcinoma In Situ; IDC = Invasive Ductal Carcinoma; F-C Chs = Fibro-cystic changes; HP = Histopathology; ILC = Invasive Lobular Carcinoma; L-A = Luminal A type; L-B = Luminal B type; NCO = Not Carried Out; T-N = Triple-Negative; NCO = Not Carried Out; NN = Non-Neoplastic lesions; NOS = Not Otherwise Specified; M = Malignancies; Mixed = IDC+ ILC; MUC = Mucinous; OP = Surgical procedure.

**Table 3 cancers-17-00039-t003:** Characterization of reported *BRCA* variants.

Gene	Cases	Gene Variant	Variant Type	ACMG Score	Associated Phenotype	Relevant Literature
*BRCA1*	3	NM_007300.4:c.5329dup p.(Gln1777ProfsTer74)	Duplication frameshift	Pathogenic (PVS1, PP5, PM2)	TN	[4,62,63,64,65]
					L-B	
					NCO	
*BRCA1*	2	NM_007300.4: c.843_846del p.(Ser282TyrfsTer15) l	Deletion, frameshift	Pathogenic (PVS1, PP5, PM2)	TN	[66,67,68]
					TN	
*BRCA1*	1	NM_007300.4: c.5093_5096del p.(Thr1698IlefsTer2)	Deletion, frameshift	Pathogenic (PVS1, PP5, PM2)	TN	[63,69,70]
*BRCA2*	1	NM_000059.4: c.2471T>G p.(Leu824Ter)	Substitution, Missense	Pathogenic (PVS1, PP5, PM2)	TN	[71,72]
*BRCA2*	1	NM_000059.4: c.5576_5579del (p.Ile1859fs)	Deletion, frameshift	Pathogenic (PVS1, PP5, PM2)	L-A	[73,74,75,76,77]
*BRCA2*	1	NM_000059.3: c.8331+1G>Ap.?	Substitution, Missense splice site, intron 18	Pathogenic (PVS1, PP5, PM2)	L-A	[78,79,80,81]

Legend: BC = Breast carcinoma; L-A = Luminal A type BC; L-B = Luminal B type BC; NCO = Not Carried Out; T-N = Triple-Negative BC.

**Table 4 cancers-17-00039-t004:** Profiles of the two subgroups of BRCA carriers.

Groups		*BRCA1* *Carriers*	*BRCA2* *Carriers*	Total
**Cases**		**6**	3	**9** **100%**
		**66.7%**	33.3%	
**Age**	** *≤40 years* **	2	0	**χ^2^ Test “*p*” value** 0.257
	** *>40 years* **	**4**	**3**	
**Family history for breast cancer**	** *Yes* **	**4**	1	**χ^2^ Test “*p*” value** 0.3428
	** *No* **	2	**2**	
**Diagnostic** **procedure**	** *CL* **	1	0	**χ^2^ Test “*p*” value** 0.569
	** *CL + BIO* **	**3**	1	
	** *OP + HP* **	2	**2**	
**Histogenetic** **type**	** *NOS* **	1	0	**Fisher’s exact test** **1**
	** *F-C Ch* **	0	0	
	** *M* **	**5**	**3**	
**Histopathological diagnosis**	** *NCO* **	1	0	**Fisher’s exact test** 0.583
	** *NN* **	0	0	
	** *DCIS* **	0	0	
	** *IDC* **	**5**	**2**	
	** *ILC* **	0	1	
**Molecular** **classification**	** *NCO* **	1	0	**χ^2^ Test “*p*” value** 0.1009
	** *L-A* **	0	**2**	
	** *L-B* **	1	0	
	** *HER2+* **	0	0	
	** *T-N* **	**4**	1	

Legend: BIO = Biopsy; CL = Clinical diagnosis; DCIS = Ductal Carcinoma In Situ; F-C Chs = Fibro-cystic changes; HP = Histopathology; IDC = Invasive Ductal Carcinoma; ILC = Invasive Lobular Carcinoma; L-A = Luminal A type; L-B = Luminal B type; M = Malignancies; NCO = Not Carried Out; NN = Non-Neoplastic lesions; NOS = Not Otherwise Specified; OP = Surgical procedure; T-N = Triple-Negative.

**Table 5 cancers-17-00039-t005:** Comparison between molecular phenotypes defined by the immuno-histochemical assays and genetic variants.

BRCA Tested Groups		Non-Carriers	*BRCA1* *Carriers*	*BRCA2* *Carriers*	χ^2^ Test “*p*” Value
**Molecular** **classification**	** *L-A* **	**12**	0	**2**	**0.0138**
	** *L-B* **	7	1	0	
	** *HER2+* **	5	0	0	
	** *T-N* **	2	**4**	1	

Legend: BIO = Biopsy; L-A = Luminal A type; L-B = Luminal B type; NCO = Not Carried Out; T-N = Triple-Negative.

## Data Availability

The original contributions presented in this study are included in the article. Further inquiries can be directed to the corresponding author.

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
