# Peer review of "Clinically Significant BRCA1 and BRCA2 Germline Variants in Breast Cancer—A Single-Center Experience"

_cancers, 2024, doi:10.3390/cancers17010039_

Round 1
Reviewer 1 Report
Comments and Suggestions for Authors
This study evaluates the prevalence of pathogenic and likely pathogenic BRCA1 and BRCA2 germline variants in breast cancer patients in a single-center cohort in Southwest Romania. Using next-generation sequencing (NGS), the authors identified distinct mutational patterns and associated phenotypes between BRCA1 and BRCA2 carriers, offering insights into genetic epidemiology and clinical outcomes. The findings provide a valuable contribution to the understanding of hereditary breast cancer in a regional context but require improvements in clarity, presentation, and grammar. Additionally, the discussion and conclusions should better contextualize the findings within global and regional research.
1. Abstract
Line 3: Rephrase “linked to pathogenic or likely pathogenic germline BRCA1 and BRCA2 variants” for conciseness, e.g., “associated with BRCA1/2 germline variants.”
Line 8: Replace “carriers of BRCA1 PVs develop a more aggressive tumor phenotype” with “BRCA1 carriers tend to develop more aggressive tumors.”
2. Introduction
Line 48: Add a citation to support “In 2020 approximately 2.3 million new breast cancer cases were reported.”Provide updated statistics on cancer as well as this cancer type prevalence, including survival rates, to highlight the critical need for prognostic biomarkers. Cite “Cancer statistics, 2024, 2024”. Then give introduction in cancer therapy in general, cite NIH paper “Cancer treatments: Past, present, and future, 2024” for more information.
Line 58: Rephrase “these apparent lower chances of survival may suggest late diagnosis” to “the lower survival rates may be attributed to late-stage diagnosis.”
Line 65: Expand on how genetic factors such as BRCA1/2 mutations contribute to therapeutic resistance and aggressive phenotypes with supporting references.
Line 82: Provide specific examples of somatic versus germline mutations and their distinct roles in breast cancer progression.
3. Materials and Methods
Study population:
Line 95: Specify whether inclusion criteria were limited to female patients or included male breast cancer cases.
Histological and molecular analysis:
Line 107: Clarify the clinical significance of the antibodies listed in Table 1, e.g., how Ki-67 indices inform prognosis.
Genetic testing:
Line 121: Explain how potential sequencing errors were minimized during NGS and manual curation.
Line 133: Provide a brief justification for using ENSEMBL VEP and how it aligns with ACMG classification standards.
4. Results
Demographics and clinical characteristics:
Line 151: Discuss why age distributions between BRCA carriers and non-carriers were not statistically significant despite biological implications.
Histopathological findings:
Line 203: Elaborate on the predominance of invasive ductal carcinoma in BRCA1 carriers compared to lobular carcinoma in BRCA2 carriers and its clinical relevance.
Variant characterization:
Line 230: Add information on the potential founder effects for identified mutations, especially BRCA1 c.5329dup.
Line 263: Clarify why some variants (e.g., BRCA2 c.2471T>G) are newly reported in Romania and whether this reflects underreporting or genuine rarity.
5. Discussion
Line 311: Discuss whether the lower incidence of VUS in this study compared to other reports is due to differences in classification criteria or sample size.
Line 326: Expand on the implications of BRCA1 mutations being more prevalent and aggressive in this cohort, particularly for regional or ethnic-specific screening protocols.
Line 350: Include comparisons with larger-scale studies from other regions to contextualize the relatively small sample size of 58 patients.
Line 375: Discuss the impact of healthcare accessibility and infrastructure on the availability and effectiveness of genetic testing in Romania.
Line 660: Suggest future studies that could validate these findings in patient-derived xenograft models or larger cohorts. Previous studies using xenograft models of breast cancer should be mentioned, such as “Comparing volatile and intravenous anesthetics in a mouse model of breast cancer metastasis, 2018”. Recently very hot single cell sequencing should also be mentioned, as it help us understand the microenvironment of the cancer cells, refer to recent single-cell sequencing studies in breast cancer and breast cacner biomarkers such as “Identification of the novel exhausted T cell CD8 + markers in breast cancer, 2024” and linke to the current study to discuss future directions.
6. Conclusion
Line 441: Rephrase “BRCA1/2 gene testing or panel testing that includes the two could be used” to “BRCA1/2 testing or broader genetic panels could be implemented.”
Line 445: Suggest actionable recommendations for integrating genetic testing into national healthcare policies, emphasizing cost-effectiveness and accessibility.
Author Response
This study evaluates the prevalence of pathogenic and likely pathogenic BRCA1 and BRCA2 germline variants in breast cancer patients in a single-center cohort in Southwest Romania. Using next-generation sequencing (NGS), the authors identified distinct mutational patterns and associated phenotypes between BRCA1 and BRCA2 carriers, offering insights into genetic epidemiology and clinical outcomes. The findings provide a valuable contribution to the understanding of hereditary breast cancer in a regional context but require improvements in clarity, presentation, and grammar. Additionally, the discussion and conclusions should better contextualize the findings within global and regional research.
- Abstract.
Line 3: Rephrase “linked to pathogenic or likely pathogenic germline BRCA1 and BRCA2 variants” for conciseness, e.g., “associated with BRCA1/2 germline variants.”
Thank you for the suggestion. The line was rephrased accordingly.
Line 8: Replace “carriers of BRCA1 PVs develop a more aggressive tumor phenotype” with “BRCA1 carriers tend to develop more aggressive tumors.”
Thank you for the suggestion. The line was rephrased accordingly.
- Introduction
Line 48: Add a citation to support “In 2020 approximately 2.3 million new breast cancer cases were reported.”Provide updated statistics on cancer as well as this cancer type prevalence, including survival rates, to highlight the critical need for prognostic biomarkers. Cite “Cancer statistics, 2024, 2024”. Then give introduction in cancer therapy in general, cite NIH paper “Cancer treatments: Past, present, and future, 2024” for more information.
Thank you for the suggestion. Both citations were added to the text along with a line that briefly addresses more recent therapeutic strategies. Therapy in breast cancer is also addressed several lines later explaining the relation between molecular subtypes and targeted therapy.
Line 58: Rephrase “these apparent lower chances of survival may suggest late diagnosis” to “the lower survival rates may be attributed to late-stage diagnosis.
Thank you for the suggestion. The line was rephrased accordingly.
Line 65: Expand on how genetic factors such as BRCA1/2 mutations contribute to therapeutic resistance and aggressive phenotypes with supporting references.
Thank you for the suggestion. We agree and introduced the paragraph between lines 110-121 along with supporting references.
Line 82: Provide specific examples of somatic versus germline mutations and their distinct roles in breast cancer progression.
We addressed this comment in paragraphs 100-103 and 110-120 along with supporting references.
- Materials and Methods
Study population:
Line 95: Specify whether inclusion criteria were limited to female patients or included male breast cancer cases.
Thank you for your suggestion. We added the phrase between 127-128.
Histological and molecular analysis:
Line 107: Clarify the clinical significance of the antibodies listed in Table 1, e.g., how Ki-67 indices inform prognosis.
Thank you for your suggestion. We added a column to Table 1 and briefly addressed the issue between lines 146-150 as the clinical significance is also addressed previously between lines 74-86.
Genetic testing:
Line 121: Explain how potential sequencing errors were minimized during NGS and manual curation.
Thank you for the suggestion. As mentioned in lines 158-160, only high coverage and depth variants were taken into consideration. We added that situationally we would use capillary sequencing to obtain the full coverage of the exons
Line 133: Provide a brief justification for using ENSEMBL VEP and how it aligns with ACMG classification standards.
Thank you for the suggestion. We used ENSEMBL VEP for annotation while filtering was done following only ACMG guidelines and classification standards. While we also had other annotation tools at our disposal and use them in case of potential discrepancies, some studies indicate that VEP is the most accurate (PMID: 36268089).
- Results
Demographics and clinical characteristics:
Line 151: Discuss why age distributions between BRCA carriers and non-carriers were not statistically significant despite biological implications.
Thank you for the suggestion. We rephrased the paragraph between 186-191 and discussed in the dedicated section, between 381-385. We found that this lack of statistical significance is encountered in other studies as well, perhaps due to late diagnosis or reporting.
Histopathological findings:
Line 203: Elaborate on the predominance of invasive ductal carcinoma in BRCA1 carriers compared to lobular carcinoma in BRCA2 carriers and its clinical relevance.
Thank you for the suggestion. We expanded this in the paragraph between 386-393
Variant characterization:
Line 230: Add information on the potential founder effects for identified mutations, especially BRCA1 c.5329dup.
Thank you for the suggestion. We added that, to indicate a potential founder effect for some of the mutations, further and broader studies are needed to better appreciate the distribution of these variants in the Romanian population.
Line 263: Clarify why some variants (e.g., BRCA2 c.2471T>G) are newly reported in Romania and whether this reflects underreporting or genuine rarity.
Thank you for your suggestion. We indeed added a paragraph between 500-507 expanding on the main reasons why some variants are newly reported in Romania.
- Discussion
Line 311: Discuss whether the lower incidence of VUS in this study compared to other reports is due to differences in classification criteria or sample size.
Thank you for your suggestion. We added a paragraph between 400-404 explaining this issue which is also linked to the above comment regarding the availability of genetic testing in our country.
Line 326: Expand on the implications of BRCA1 mutations being more prevalent and aggressive in this cohort, particularly for regional or ethnic-specific screening protocols.
Thank you for your comment. In essence, the same causality and context applies as explained in the previous two comments.
Line 350: Include comparisons with larger-scale studies from other regions to contextualize the relatively small sample size of 58 patients.
Thank you for your suggestion. We agree and addressed this in paragraphs 381-393 and 400-404.
Line 375: Discuss the impact of healthcare accessibility and infrastructure on the availability and effectiveness of genetic testing in Romania.
Thank you for the suggestion. We agree with the issue and expanded this in the paragraph between 500-507
Line 660: Suggest future studies that could validate these findings in patient-derived xenograft models or larger cohorts. Previous studies using xenograft models of breast cancer should be mentioned, such as “Comparing volatile and intravenous anesthetics in a mouse model of breast cancer metastasis, 2018”. Recently very hot single cell sequencing should also be mentioned, as it help us understand the microenvironment of the cancer cells, refer to recent single-cell sequencing studies in breast cancer and breast cacner biomarkers such as “Identification of the novel exhausted T cell CD8 + markers in breast cancer, 2024” and linke to the current study to discuss future directions.
Thank you for the suggestion. We incorporated the paragraph between 518-523 to also address such studies.
- Conclusion
Line 441: Rephrase “BRCA1/2 gene testing or panel testing that includes the two could be used” to “BRCA1/2 testing or broader genetic panels could be implemented.”
Thank you for the suggestion. The line was rephrased accordingly.
Line 445: Suggest actionable recommendations for integrating genetic testing into national healthcare policies, emphasizing cost-effectiveness and accessibility.
Thank you for the suggestion. We added a few words in the conclusion to try and cover the issue as low accessibility is mainly due to a difficult accommodation of the general population to such tests, especially when it comes to individuals coming from rural areas of the country.
Reviewer 2 Report
Comments and Suggestions for Authors
It is a nice work to show the possible relationship between tumor phenotypes and germline variants of BRCA1/2 gene. The data provided by this work is overall complete and convincing. Some minor modifications can be made for the better improvement.
1. A simple experimental procedure can be made to visualize the performance and analysis.
2. A scale bar can be added in each figures from Figure 1. furthermore, a semi-quantification of the positive areas can be also performed to make a column graph.
3. There is seemingly no direct comparison between molecular phenotyes from the immuno-histochemical assays and genetic variants. The molecular phenotyes and all genetic variants referred in this paper can be also added in Table 3.
Author Response
It is a nice work to show the possible relationship between tumor phenotypes and germline variants of BRCA1/2 gene. The data provided by this work is overall complete and convincing. Some minor modifications can be made for the better improvement.
- A simple experimental procedure can be made to visualize the performance and analysis.
Thank you for your suggestion. We highlighted the main steps of the study in the Materials and Methods so they can be better visualized.
- A scale bar can be added in each figures from Figure 1. furthermore, a semi-quantification of the positive areas can be also performed to make a column graph.
Thank you for your suggestion. We agree and have made modifications between lines 246-265
- There is seemingly no direct comparison between molecular phenotypes from the immuno-histochemical assays and genetic variants. The molecular phenotypes and all genetic variants referred in this paper can be also added in Table 3.
Thank you for your remark and suggestions. We added the column in Table 3 and expanded on this in a new section between lines 341-356
Reviewer 3 Report
Comments and Suggestions for Authors
The study “BRCA1 and BRCA2 clinically significant germline variants in 2 breast cancer—Single center experience” by Plesea, R.M. et al., evaluated the germline variants in the BRCA1 and BRCA2 genes in patients with a family history of breast cancer. Authors have reported that 15.5% of 58 patients studied in this investigation, have carried either BRCA1 or BRCA2 Pathogenic (PV) or Likely 34 Pathogenic variants (LPV). BRCA1 carriers had aggressive tumors whereas BRCA2 carriers had ra-35 ther low-grade tumors. Authors have concluded that carriers of BRCA1 PVs develop a 37 more aggressive tumor phenotype than carriers of BRCA2 PVs and patients with no germline PVs in either of the two genes.
Comments
Line #55 and 57: Correct the numbers 33%000, and 23%000 and 27%000
The sample size is very low. It would have been more convincing if the sample size is increased as well as studied in multiple centers
Authors have to highlight the significance of this study and mention how the results can impact the existing treatment procedures.
Since the variants of BRCA1 and BRCA2 are thoroughly studied in breast cancers, as such the study is an additional information providing investigation. Hence, it is important to highlight how the results of this study helps in making any treatment decisions.
Author Response
The study “BRCA1 and BRCA2 clinically significant germline variants in 2 breast cancer—Single center experience” by Plesea, R.M. et al., evaluated the germline variants in the BRCA1 and BRCA2 genes in patients with a family history of breast cancer. Authors have reported that 15.5% of 58 patients studied in this investigation, have carried either BRCA1 or BRCA2 Pathogenic (PV) or Likely 34 Pathogenic variants (LPV). BRCA1 carriers had aggressive tumors whereas BRCA2 carriers had ra-35 ther low-grade tumors. Authors have concluded that carriers of BRCA1 PVs develop a 37 more aggressive tumor phenotype than carriers of BRCA2 PVs and patients with no germline PVs in either of the two genes.
Comments
Line #55 and 57: Correct the numbers 33%000, and 23%000 and 27%000
Thank you for your suggestion. We agree and have made correction.
The sample size is very low. It would have been more convincing if the sample size is increased as well as studied in multiple centers
Thank you for your suggestion. We agreed and added paragraphs across the text explaining the small sample size.
Authors have to highlight the significance of this study and mention how the results can impact the existing treatment procedures.
Thank you for your suggestion. We added a phrase to address this in the Conclusion section.
Since the variants of BRCA1 and BRCA2 are thoroughly studied in breast cancers, as such the study is an additional information providing investigation. Hence, it is important to highlight how the results of this study helps in making any treatment decisions.
Thank you for your remark. We highlighted the impact on treatment decisions in Section 4.3, lines 452-464
Round 2
Reviewer 1 Report
Comments and Suggestions for Authors
good